# OpenReview forum: "GeoDM: Geometry-aware Distribution Matching for Dataset Distillation"
_ICML.cc/2026/Conference — ICML 2026 regular_

### Official Review · Reviewer_LbgH · 2026-03-03

**Soundness:** 3
**Presentation:** 3
**Significance:** 2
**Originality:** 2
**Overall Recommendation:** 5
**Confidence:** 3

**Summary:**

This paper presents a novel geometry-aware distribution-matching framework for dataset distillation task, which captures the distribution of real data by modeling the real data distribution on three types of constant-curvature manifolds, i.e., Euclidean, hyperbolic and spherical, to help synthetic data better align with the real data. To further improve the faithfulness of the distilled dataset, the paper introduces an auxiliary OT loss to help align the synthetic and real data distributions in the product-manifold embedding space. The paper also provides a series of theoretical analyses to justify the geometric design, including a geometry-driven risk decomposition and a proof that product manifolds yield strictly tighter generalization bounds than Euclidean matching alone under mild regularity assumptions. Empirical results on standard benchmarks demonstrate that the proposed framework achieves superior performance compared to prior dataset distillation methods, while maintaining robustness under various distribution-matching strategies and single-geometry variants.

**Compliance With Llm Reviewing Policy:**

Affirmed.

**Final Justification:**

Thanks to the authors for the updates. I think the authors have provided enough reasoning and empirical evidence to resolve my concerns, so I have increased my score.

**Key Questions For Authors:**

1. This work extends NCFM, where NCFM performs distribution matching based on the Neural Characteristic Function, whereas the current work matches distributions in Riemannian space. However, the paper directly adopts the min-max loss from NCFM without comparing how the use of Riemannian geometry affects the ability to capture dataset distributions. Have the authors conducted any investigation or analysis regarding this comparison?
2. GeoDM performs distribution matching across three types of Riemannian spaces. However, the paper provides limited discussion on the individual contributions of each space. Have the authors analyzed the relative weights (α, β, γ) assigned to the three spaces in practice, and which space contributes most to aligning the distilled data with the real distribution? Additionally, since the product space combines multiple spaces and thus has a higher total embedding dimension than any single space, have the authors conducted experiments controlling for total dimensionality to verify that the observed improvements are not simply due to the increased dimensionality? Addressing these questions could help better understand the principles underlying GeoDM.
3. How are the feature extractors for the different geometric spaces obtained? Do they require additional training, or are they shared/reused from existing models?
4. What is the specific contribution of the auxiliary OT loss to GeoDM? Could the authors clarify whether its effectiveness is unique to the proposed GeoDM framework, or if it can be considered a generally applicable optimization technique for distribution matching?

**Limitations:**

yes

**Strengths And Weaknesses:**

**Strengths**:
1. Theory: The discussion on why dataset distillation benefits from incorporating non-Euclidean geometry and why modeling data in a product space improves performance is largely self-consistent and theoretically well-motivated.
2. Experiments: The paper includes sufficiently comprehensive baseline comparisons and ablation studies. The empirical results generally support the main claims and validate the proposed design.
3. Presentation: The paper is well organized, with clear and detailed descriptions of both the methodology and the theoretical analysis.

**Major Weaknesses**:
1. Limited originality: The methodological contribution appears incremental and somewhat compositional. As discussed by the authors in Appendix F, prior work on dataset distillation has already explored the use of hyperbolic space within the Riemannian manifold framework for distribution matching, which reduces the degree of novelty of the current approach.
2. Additional computational overhead: To extract embeddings from three different geometric spaces, the framework requires multiple feature extractors, which introduces additional computational and memory overhead compared to single-geometry approaches.
3. Limited benchmark coverage: The empirical evaluation is conducted primarily on relatively simple benchmarks (i.e., MNIST, CIFAR-10, and CIFAR-100). More challenging and commonly used dataset distillation benchmarks, such as Tiny-ImageNet, are not included, which limits the assessment of the method’s scalability and generalization to more complex scenarios.
4. Justification of the OT loss: The introduction of the auxiliary OT loss appears somewhat abrupt. While the authors claim that it improves the “representational faithfulness of the distilled dataset,” they also emphasize that it is a “complementary and non-essential component” of the framework. This dual positioning makes its necessity somewhat unclear. Moreover, based on the reported results, the additional computational overhead introduced by the OT loss may not be fully justified by the performance gains. The paper also does not discuss whether incorporating OT constitutes a generally applicable enhancement for distribution matching, or if its effectiveness is specific to the proposed setting.

**Minor Weaknesses**:
1. Formatting issue: Table 6 appears to be missing a column header.

---

> ### Author Rebuttal · Authors · 2026-03-31
>
> **Weakness 1**: Please refer to the response to Reviewer iyTn in Question 1 for experimental results and response to Revewier csNB in Weakness 1 for originality.
>
> **Weakness 2**: We acknowledge that our framework introduces additional computational overhead. However, we note that the total embedding dimensionality is kept the same as NCFM, ensuring a fair comparison in representation capacity.
>
> Importantly, our computational analysis (Table 3) shows that, for the same target performance, GeoDM requires fewer training epochs and lower total FLOPs compared to NCFM.
> In addition, Appendix E.1 (Table 6) provides a detailed breakdown of the computational and memory overhead of each component, along with their corresponding performance gains. These results demonstrate a clear trade-off between computation and accuracy, which we believe is well justified given the consistent performance improvements.
>
> **Weakness 3**: We further include experiments on the more challenging Tiny-ImageNet benchmark. As shown in Table belowm GeoDM consistently outperforms strong baselines (DM, MTT, and NCFM) across all IPC settings.
>
> | Method | IPC=1 | IPC=10 | IPC=50 |
> |----|----|----|----|
> | DM | 3.9±0.2  | 12.9±0.4 | 24.1±0.3 |
> | MTT | 8.8±0.3  | 23.2±0.2 | 28.0±0.3 |
> | NCFM | 18.2±0.5 | 26.8±0.6 | 29.6±0.5 |
> | GeoDM | **19.4±0.4** | **27.1±0.5** | **30.5±0.7** |
>
> **Weakness 4**: We clarify that the OT loss is designed as a refinement module rather than a core component of GeoDM. The main contribution of our method lies in geometry-aware distribution matching in the product space, while OT further improves alignment fidelity across different geometric components. This is consistent with our ablation results (Table 5), where the product space alone already provides strong gains, and OT brings additional improvements.
>
> Regarding computational cost, we refer the reviewer to Table 7 for a detailed analysis of the OT loss. While OT introduces additional overhead, the results show that it consistently improves performance with a moderate increase in computation.
>
> Finally, OT has been explored in prior dataset distillation works, and our contribution is to integrate it into a geometry-aware setting, where it helps improve the fidelity of multiple geometries during distribution matching.
>
> **Minor Weakness**: Thanks for pointing out, we will revise in the final version.
>
> **Question 1**: We isolate the effect of geometry by replacing NCFM with alternative DM methods (Fig. 3). The consistent improvements show that the gain comes from the geometry-aware representation rather than the min-max loss.
> From a theoretical perspective, NCFM depends on the embedding space. Euclidean embeddings distort hierarchical or angular structures, while our product manifold with adaptive curvature reduces such distortion (Theorem 4.1), leading to more faithful distribution matching.
>
> **Question 2**: First, regarding the contribution of different geometric spaces, we have provided empirical analysis in the appendix (Table 10), where we evaluate different space combinations. The results show that using only a subset of geometries leads to limited improvement, while combining Euclidean, hyperbolic, and spherical spaces consistently achieves the best performance. This indicates that each space captures complementary structural properties of the data.
>
> We further analyze the role of the geometry weights by conducting additional experiments where one of the weights is fixed. As shown in Table below, fixing any single geometry weight leads to a consistent drop in performance compared to the fully learnable setting, although all variants still outperform the baseline. This indicates that different geometric components contribute complementary information, and adaptively balancing their contributions is important for optimal performance.
>
> | Method | Cifar10-IPC10 |
> |----|---|
> | NCFM | 71.8 |
> | GeoDM (fixed weight E) | 73.9 |
> | GeoDM (fixed weight S) | 73.6 |
> | GeoDM (fixed weight H) | 73.3 |
> | GeoDM | **74.4** |
>
> Second, regarding the concern about dimensionality, we explicitly control for this factor in our experimental setup. The total embedding dimension of GeoDM is kept the same as NCFM (see line 304), ensuring a fair comparison. In addition, Appendix Table 8 provides further analysis of different dimensionalities, showing that performance is stable across a range of dimensions and not driven by increased capacity.
>
> **Question 3**:The feature extractors are implemented within a single network, where Euclidean, hyperbolic, and spherical branches share the same backbone and are jointly trained end-to-end, without additional pretraining.
> The model extracts features using a standard backbone and maps them into different geometric spaces via geometry-aware operations. Learnable curvature and weights are optimized jointly, ensuring a fair comparison without extra models or external data.
>
> **Question 4**: Please refer to the response to Weakness 4 in this rebuttal.

---

> > ### Author Rebuttal · Reviewer_LbgH · 2026-04-03
> >
> > I think the authors have provided enough reasoning and empirical evidence to resolve my concerns.

---

> > > ### Author Response · Authors · 2026-04-04
> > >
> > > Thank you for your dedicated time and for raising the score. Your support is much appreciated.

---

### Official Review · Reviewer_iyTn · 2026-03-07

**Soundness:** 3
**Presentation:** 2
**Significance:** 3
**Originality:** 3
**Overall Recommendation:** 4
**Confidence:** 5

**Summary:**

This paper studies dataset distillation from a geometry-aware distribution matching perspective. The main idea is that existing distribution matching methods largely operate in Euclidean feature space and may therefore fail to capture the mixed-curvature structure of real data. To address this, the paper proposes GeoDM, which performs matching in a product manifold composed of Euclidean, hyperbolic, and spherical components, with learnable curvature parameters, learnable geometry weights, and a geometry-aware OT term. The paper also presents a generalization analysis arguing that the product-space formulation can yield a tighter error bound than a purely Euclidean formulation, and reports experiments on MNIST, CIFAR-10, CIFAR-100, Imagenette, and ImageWoof.

**Compliance With Llm Reviewing Policy:**

Affirmed.

**Final Justification:**

Based on the rebuttal, I find that most of my major concerns have been adequately addressed. There are still a small number of remaining issues, but overall, I am willing to raise my score to 4 (Weak Accept).

The main remaining concerns are as follows:

Methods such as D4M and IGD were not specifically designed for small-scale datasets like CIFAR. Therefore, to make the comparison more convincing, the paper should also include experiments on relatively larger datasets, such as ImageNette and ImageWoof.
The evidence for generalization to larger-scale datasets and across different backbone architectures is still limited. As such, the claimed scalability and generalization ability of the proposed method are not yet fully validated.

**Key Questions For Authors:**

1. The paper cites closely related geometry-oriented distillation work, including Hyperbolic Dataset Distillation. However, the main experiments do not include a direct comparison with such methods. Why is this comparison missing? Where do the authors expect GeoDM to outperform the most relevant geometry-based baselines?

2. The theoretical claims are an important part of the paper. In particular, the tighter-bound argument for product space plays a central role. Can the authors provide empirical analyses that connect more directly to the theory? For example, can they report distortion-related proxies, manifold fidelity measures, or class-structure preservation metrics for Euclidean and product-space embeddings?

3. The method uses a fixed three-geometry decomposition and a specific dimensional split. How sensitive are the results to these choices? What happens if the method uses only two geometries, different dimensional allocations, or fixed curvature instead of learnable curvature?

4. Can the authors provide stronger evidence that the gains do not depend heavily on ConvNet-3 and the current benchmark scale? Additional results on stronger backbones, larger budgets, or more challenging datasets would make the paper much stronger.

**Limitations:**

Yes

**Strengths And Weaknesses:**

Strengths:
The paper addresses a meaningful problem. It is reasonable to question whether Euclidean distribution matching is sufficient for dataset distillation, and the motivation that real data may exhibit heterogeneous geometric structure is at least intuitively plausible. The proposed method is also more than a trivial metric replacement: it combines a product-space representation, learnable curvature, learnable geometry weights, and an OT-based alignment term within a unified distillation framework. Empirically, the method shows improvements over several prior baselines on CIFAR-10 and CIFAR-100, and the paper includes cross-architecture evaluation, higher-resolution experiments, and component ablations.

Weaknesses:
1. The theoretical section makes strong claims about why product-space geometry should outperform Euclidean space. However, these claims depend on assumptions that already favor the proposed method. The paper also does not provide enough empirical evidence that directly tests the theoretical intuition. As a result, the theory feels somewhat detached from the experimental section.

2. The current ablations show that the full method performs better than its partial variants. However, they do not clearly prove that mixed-curvature modeling is the main reason for the gains. The paper also does not justify several important design choices well. These choices include the use of exactly three geometries, the dimensional split, and the learnable curvature design.

3. The paper evaluates on several datasets, but most results still come from relatively standard and modest-scale settings. I would expect stronger validation in larger-scale or more modern settings.

4. The experimental comparison omits recent diffusion-based generative distillation methods such as Minimax [1] and D⁴M [2]. These relevant baselines are neither discussed in sufficient detail nor included in the empirical evaluation.

5. Some sentences in the main text are awkward or unclear. For example, the experimental setup section includes the sentence “Without explicit evidence, Our experiments are conducted on 3 spaces across all setting.” This sentence is difficult to parse and is not grammatically correct. The tables also contain formatting issues, such as the entry “49, 4,” which appears to be a clear typo. In addition, some claims are stated too strongly relative to the reported gains. For instance, the paper says that “these results demonstrate more than incremental improvements,” but many of the improvements look steady yet still fairly modest in magnitude.

[1] Efficient Dataset Distillation via Minimax Diffusion

[2] D⁴M: Dataset Distillation via Disentangled Diffusion Model

---

> ### Author Rebuttal · Authors · 2026-03-31
>
> **Weakness 1**: We clarify that our theory does not assume data strictly lie in a product space, but isolates the effect of geometric distortion in distribution matching. The key result is that product spaces reduce distortion and yield tighter bounds when data exhibit mixed geometries.
>
> This is supported empirically: Table 5 shows that the product space alone improves performance, and Figure 3 demonstrates consistent gains across different objectives, aligning with our theoretical intuition.
>
> **Weakness 2**: Regarding whether the gains mainly come from mixed-curvature modeling, our ablation study (Table 5) isolates this effect: the product space alone, without curvature learning or OT, already improves performance over single-geometry baselines by $1.7\%$, indicating that mixed-curvature modeling is the primary contributor.
> For the design choices, you can refer to the response to Question 3 in this rebuttal.
>
> **Weakness 3**: In addition to standard benchmarks, we evaluate our method on higher-resolution and more challenging datasets, including Imagenette, ImageWoof, and ImageNet-100 (Table 4 and Table 12). These settings involve more complex data distributions and larger input scales compared to CIFAR.
>
> We further conducted experiments on tiny-imagenet. The results can be seen in the response to Reviewer LbgH in weakness 3.
>
> **Weakness 4**: We include additional comparisons with recent diffusion-based distillation methods (Minimax, D4M, and DiT-IGD). As shown in Table below, GeoDM consistently outperforms these approaches across datasets and IPC settings.
>
> | Method | Cifar10-IPC10 | Cifar10-IPC50 | Cifar100-IPC10 | Cifar100-IPC50 |
> |--------|--------|--------|---------|---------|
> | Minimax | 45.1±0.9 | 54.5±0.7 | 31.2±0.4 | 38.6±0.7 |
> | D4M     | 56.2±0.4 | 72.8±0.5 | 45.0±0.1 | 48.8±0.3 |
> | DiT-IGD | 54.8±0.5 | 66.8±0.5 | 45.8±0.5 | 53.9±0.6 |
> | GeoDM   | **74.4±0.3** | **78.3±0.2** | **49.2±0.3** | **55.0±0.2** |
>
> **Weakness 5**: We will revise the manuscript to improve clarity and correctness. Thank you for your suggestions.
>
> **Question 1**: We further include comparisons with hyperbolic dataset distillation methods (HDD). As shown in Table below, while HDD improves over Euclidean baselines by incorporating non-Euclidean geometry, GeoDM consistently achieves superior performance across all datasets and IPC settings. This demonstrates the advantage of modeling mixed geometries in a product space rather than relying on a single hyperbolic space.
>
> | Data | DM-HDD | IDM-HDD | GeoDM |
> |----|----|------|----|
> | MNIST-1   | 90.3±0.5 | 95.4±0.7 | **96.1±0.2** |
> | MNIST-10  | 96.8±0.5 | 97.3±0.2 | **98.2±0.2** |
> | MNIST-50  | 97.3±0.3   | 98.1±0.3 | **99.2±0.1** |
> | CIFAR10-1     | 26.9±0.6 | 45.7±0.6 | **51.2±0.2** |
> | CIFAR10-10    | 49.8±0.5 | 60.7±0.7 | **74.4±0.3** |
> | CIFAR10-50    | 61.6±0.3 | 69.9±0.3 | **78.3±0.2** |
> | CIFAR100-1    | 13.0±0.4   | 26.2±0.4 | **38.0±0.4** |
> | CIFAR100-10   | 30.1±0.3   | 44.1±0.5 | **49.2±0.3** |
> | CIFAR100-50   | 43.6±0.3   | 48.6±0.6 | **55.0±0.2** |
>
> **Question 2**: While we do not explicitly compute distortion metrics, we provide empirical evidence that directly reflects the role of geometry in our formulation. Specifically, we conduct an experiment where 50% of CIFAR-10 data is replaced with Gaussian noise. You can refer to the response to Reviewer hvXa in Question 3.
>
> **Question 3**: We have provided detailed empirical analysis of these design choices in the appendix. In Appendix E.2 (Dimensionality Analysis), we study how different total dimensions affect performance. Table 8 shows that the dimensionality used in our paper is selected based on the best performance from a grid search. In Table 9, we further analyze different dimension allocation strategies across spaces. The results show that varying the allocation does not significantly affect performance, although Euclidean-dominant allocations tend to better align with dataset distillation objectives.
>
> In Appendix E.3, we further evaluate different combinations of geometric spaces. We observe that using only two geometries does not consistently improve performance, while the full product space with all three geometries achieves the best results, suggesting that combining multiple geometries is necessary to capture diverse structural properties in the data.
>
> We further evaluate the effect of curvature design, as shown below. Compared to the product space alone (73.5\%), fixing either spherical or hyperbolic curvature as 1 leads to slightly lower performance (73.6\% and 73.7\%), while fully learnable curvature achieves the best result (74.4\%).
>
> | Method | Cifar10-IPC10 |
> |---|---|
> | NCFM | 71.8 |
> | GeoDM (only product) | 73.5 |
> | GeoDM (fixed curvature S) | 73.6 |
> | GeoDM (fixed curvature H) | 73.7 |
> | GeoDM | **74.4** |
>
> **Question 4**: Please refer to the response to Reviewer hvXa in Question 2 for more backbones and and response to Reviewer LbgH  in Weakness 3 for more datasets.

---

> > ### Author Rebuttal · Reviewer_iyTn · 2026-04-02
> >
> > Based on the rebuttal, I find that most of my major concerns have been adequately addressed. There are still a small number of remaining issues, but overall I am willing to raise my score to 4 (Weak Accept).
> >
> > The main remaining concerns are as follows:
> >
> > Methods such as D4M and IGD were not specifically designed for small-scale datasets like CIFAR. Therefore, to make the comparison more convincing, the paper should also include experiments on relatively larger datasets, such as ImageNette and ImageWoof.
> > The evidence for generalization to larger-scale datasets and across different backbone architectures is still limited. As such, the claimed scalability and generalization ability of the proposed method are not yet fully validated.

---

> > > ### Author Response · Authors · 2026-04-02
> > >
> > > We sincerely thank the reviewer for raising the score after our previous clarification.
> > >
> > > Following the reviewer’s comment on evaluating larger-scale datasets, we conduct additional experiments on ImageNette and ImageWoof. The results are summarized in Table below.
> > >
> > > | Dataset    | IPC | Minimax | D4M  | DiT-IGD | NCFM | GeoDM |
> > > |------------|-----|---------|------|---------|------|-------|
> > > | Imagenette | 1   | 27.2    | 24.8 | 26.8    | 53.4   | **54.8** |
> > > |            | 10  | 41.2    | 38.8 | 55.0    | 77.3   | **78.8** |
> > > | ImageWoof  | 1   | 18.2    | 16.8 | 19.2    | 27.2   | **28.1** |
> > > |            | 10  | 21.6    | 19.4 | 28.6    | 47.9   | **49.1** |
> > >
> > > We observe that GeoDM consistently outperforms diffusion-based methods across both datasets and IPC settings, demonstrating strong generalization to larger-scale datasets.

---

### Official Review · Reviewer_hvXa · 2026-03-08

**Soundness:** 3
**Presentation:** 4
**Significance:** 3
**Originality:** 3
**Overall Recommendation:** 5
**Confidence:** 4

**Summary:**

This paper proposes a geometry-aware distribution matching framework for dataset distillation. Previous methods operate exclusively in Euclidean space, which does not capture curvature and other non-Euclidean structures often present in real data. GeoDM addresses this by constructing a product manifold $\mathcal{P} = \mathbb{E}^{d_E} \times \mathbb{H}^{d_H}_{c_H} \times \mathbb{S}^{d_S}_{c_S}$, combining Euclidean, hyperbolic and spherical spaces. Theoretical validation and range of empirical experiments are performed.

**Compliance With Llm Reviewing Policy:**

Affirmed.

**Final Justification:**

I like this paper overall and find the direction promising. The writing and motivation are clear. Although the method may be compositional in nature, the introduction of product Riemannian space (specifically for data distillation) is out of the box. The authors addressed my rebuttal points, I therefore updated overall score from 4 to 5.

**Key Questions For Authors:**

1. How do you see the core idea of GeoDM (creation of a product space) port to non-vision (or multi-modal) regimes? Would the underlying theory hold and only require slight modifications to the architecture design?
2. What is the sensitivity to the choice of backbone for computing the geometric embedding? Currently only ConvNet and ResNet models are tested. What about other residual architectures (EfficientNet, MobileNet) or transformer backbones (Swin, ViT)?
3. The learned curvatures converge to similar values across initializations (Table 13). Do these values differ meaningfully across datasets (e.g., C-100, Imagenette)? And can these learned curvatures provide interpretable insights into the geometry of different datasets?

Overall, I like this paper and find the direction promising. I'm currently a marginal accept but am willing to increase my score if the authors address my concerns, mainly related to reproducibility.

**Limitations:**

Yes

**Strengths And Weaknesses:**

**Strengths:**
- The motivation that distillation should respect the geometry of data manifolds and that Euclidean embeddings alone aren't sufficient for capturing the underling manifold structure is intuitive and interesting.
- Designing the system such that if incorporates learnable curvature and weights is very useful from a practicality standpoint
- The theoretical contributions are well designed. Decomposing the generalization gap into statistical, stability and geometry terms (Theorem 4.1) is quite clean. And Theorem 4.2 formalizes reduced distortion of product space relative to Euclidean space. Supporting lemmas in the appendix are also useful.
- Good performance across vision datasets compared to a wide range of baseline methods on small image datasets 32x32 (Table 1) and larger-scale images 128x128 (Table 4). The cross-architecture investigation (Table 2) proves robustness to changing embedding spaces. The isolation ablations (Figure 3) demonstrate the impact of the combined metric space compared to each in isolation.

**Weaknesses:**
- My main concern is regarding reproducibility. Algorithm 1 references three network components (standard CNN, Riemannian CNN with Poincare operations and a spherical CNN) but provides little detail on each architecture and how geometric operations (exponential maps, spherical convolutions, etc.) are concretely implemented.. The literature on geometric learning is vast (e.g., [1, 2, 3] among others) and its not obvious which specific choices are made. Both (1) more detailed explanations on these three components and (2) code, would substantially improve this reproducibility concern.
- Allowing the curvature and weights to be learnt, but keeping the dimensionality of each space ($d_E, d_H, d_S$) fixed seems inconsistent. Perhaps, having the dimensionality learnable is too much of an over-kill and essentially becomes a neural architecture search (NAS) problem. But a bit more rigor on the selection (or ratio) between these spaces would be warranted. For example, there could be some intrinsic dimension "pre-processing" which could allow better selection of these dimensions w.r.t. a dataset.
- The evaluations are limited to vision benchmarks. No NLP or other modalities are tested which weakens the generality. It's unclear how well the geometric benefits transfer to datasets with fundamentally different structure, where perhaps non-Euclidean geometry is even more natural. Although this would require changing the underlying feature extractors, as they are convolutional in nature.

---
**References:** \
[1] Ganea et al., "Hyperbolic neural networks". _NeurIPS_, 2018 \
[2] Katsman et al., "Riemannian residual neural networks." _NeurIPS_, 2023 \
[3] Masci et al., "Geodesic convolutional neural networks on Riemannian manifolds." _CVPR Workshops_, 2015

---

> ### Author Rebuttal · Authors · 2026-03-31
>
> **Weakness 1**: We provide more detailed implementation of our Riemannian CNN to improve reproducibility.
>
> - **Feature extraction.**
> Given an input $x$, we first extract convolutional features:
> $$
> h = f_{\theta}(x), \quad z = \mathrm{vec}(h) \in \mathbb{R}^d,
> $$
> where $f_{\theta}$ is a standard CNN and $\mathrm{vec}(\cdot)$ denotes flattening.
>
> - **Hyperbolic mapping.**
> For hyperbolic space (Poincaré ball with curvature $k<0$), we map $z$ using the exponential map at the origin:
> $$
> \exp_0^{\mathbb{H}}(z) = \tanh\left(\sqrt{|k|}\,\|z\|\right)\frac{z}{\|z\|}.
> $$
> The mapped point is projected to ensure $\|x\| < 1/\sqrt{|k|}$:
> $$
> \Pi(x) = \frac{x}{\max\left(1, \frac{\|x\|}{1/\sqrt{|k|}-\epsilon}\right)}.
> $$
>
> - **Spherical mapping.**
> For spherical space ($k>0$), we use:
> $$
> \exp_0^{\mathbb{S}}(z) = \sin(\|z\|)\frac{z}{\|z\|}, \quad
> \Pi(x) = \frac{x}{\|x\|}.
> $$
>
> - **Euclidean case.**
> For Euclidean space, no transformation is applied:
> $$
> z_{\mathbb{E}} = z.
> $$
>
> - **Geodesic distance.**
> The hyperbolic distance is:
> $$
> d_{\mathbb{H}}(x,y) = \frac{2}{\sqrt{|k|}} \operatorname{atanh}\left(\sqrt{|k|}\|x-y\|\right),
> $$
> and the spherical distance is:
> $$
> d_{\mathbb{S}}(x,y) = \arccos(\langle x,y\rangle).
> $$
>
> - **Mobius linear layer.**
> For hyperbolic classification, we use Möbius linear transformation:
> $$
> W \otimes x = \frac{Wx}{1 + k\|x\|^2}, \quad
> x \oplus b = \frac{x + b}{1 + k\langle x,b\rangle}.
> $$
> The output is projected back to the manifold:
> $$
> \hat{y} = \Pi\big( (W \otimes x) \oplus b \big).
> $$
>
> - **Summary.**
> Overall, the embedding process is:
> $$
> z_{\mathcal{M}} = \Pi_{\mathcal{M}} \big( \exp_0^{\mathcal{M}} ( f_{\theta}(x) ) \big),
> $$
> where $\mathcal{M} \in \{\mathbb{E}, \mathbb{H}, \mathbb{S}\}$.
> This design follows standard Riemannian operations (expmap, logmap, projection, geodesic distance).
>
> **Weakness 2**: We intentionally keep the dimensionality $(d_E, d_H, d_S)$ fixed, as making it learnable would effectively turn the problem into architecture search, introducing additional instability and complexity beyond our focus on geometry-aware matching.  While intrinsic dimension estimation could be explored, we find that a simple fixed allocation already achieves strong and robust performance, and we leave adaptive dimension selection for future work.
>
> **Weakness 3**: Extending to NLP is non-trivial, mainly due to the discrete nature of text and the need for suitable feature extractors, which makes direct optimization more challenging. However, our formulation and theoretical analysis, which highlight the role of geometry in distribution alignment, remain applicable, and we believe adapting GeoDM to NLP or multi-modal settings is a promising direction for future work.
>
> **Question 1**: Our theoretical analysis operates on embedding distributions rather than raw inputs, and thus is not tied to vision data. Theorem 4.1–4.2 show that product spaces reduce geometric distortion and yield tighter bounds, independent of modality.
> Although NLP data are discrete, they are mapped into continuous embedding spaces (e.g., via transformers), where distribution matching applies. Therefore, our formulation remains applicable given suitable feature representations.
>
> **Question 2**: We further use Transformer-12 for computing the geometric embedding and evaluate with ConvNet-3 and Transformer-12 respectively. As shown in Table, GeoDM consistently outperforms NCFM under both backbones.
> These results indicate that our method is robust across different architectures.
>
> | Method| ConvNet-3 (IPC=1) | ConvNet-3 (IPC=10) | Transformer-12 (IPC=1) | Transformer-12 (IPC=10) |
> |-----|-------|-------|-------|------|
> | NCFM  | 35.8 ± 0.7  | 54.7 ± 0.6   | 23.9 ± 0.6    | 29.0 ± 0.5    |
> | GeoDM (Ours)   | **38.2 ± 0.4**   | **56.5 ± 0.5**     | **25.1 ± 0.4**   | **30.6 ± 0.4**  |
>
> **Question 3**: We observe that the learned curvatures differ meaningfully across datasets. As shown in Table below, MNIST exhibits relatively small curvature values, while CIFAR-10 shows stronger spherical and moderate hyperbolic components, and CIFAR-100 presents more pronounced hyperbolic curvature. This suggests that the model adapts to different intrinsic geometric structures in the data.
>
> To further investigate interpretability, we conduct an additional experiment by replacing 50\% of the CIFAR-10 dataset with synthetic Gaussian noise samples in the last row of the table, which preserve the input dimensionality (32$\times$32$\times$3) but remove structured geometric patterns. In this setting, both spherical and hyperbolic curvatures decrease significantly, indicating that the learned curvature reflects the presence of geometric information in the dataset.
>
> | Dataset   | Curv.E | Curv.S  | Curv.H   |
> |-------|---|----|-----|
> | MNIST   | 0 | 0.17 | -0.135 |
> | CIFAR-10  | 0 | 1.41 | -0.79 |
> | CIFAR-100  | 0 | 0.10 | -0.98 |
> | CIFAR-10 (noise) | 0 | 0.84 | -0.48 |

---

> > ### Author Rebuttal · Reviewer_hvXa · 2026-04-02
> >
> > The authors have answered my questions appropriately and thus I'm happy to increase my score to 5.
> >
> > I still advise the authors to provide reproducible code in the future. The mathematical description of the Riemannian CNN pipeline is appreciated, but full reproducibility will require runnable code, especially given the subtleties of numerical stability in hyperbolic operations (projection clipping, gradient handling near the boundary of the Poincaré ball, etc.)

---

> > > ### Author Response · Authors · 2026-04-03
> > >
> > > We are sincerely grateful for the reviewer's positive feedback and for the time dedicated to review our paper. The code will be released once it get accepted.

---

### Official Review · Reviewer_csNB · 2026-03-12

**Soundness:** 2
**Presentation:** 2
**Significance:** 2
**Originality:** 1
**Overall Recommendation:** 4
**Confidence:** 4

**Summary:**

This paper proposes GeoDM, a geometry-aware distribution matching framework for dataset distillation. The core idea is to perform distribution matching between real and synthetic data in a Cartesian product space comprising Euclidean, hyperbolic, and spherical manifolds. Real and synthetic samples are embedded into this product space via a Riemannian convolutional neural network. On top of this product space embedding, the authors introduce several training mechanisms: learnable curvature parameters for the hyperbolic and spherical components, learnable weights that control the relative contribution of each geometry via softmax normalization, and a geometry-aware optimal transport loss that measures distributional alignment across the product manifold. The paper provides theoretical analysis arguing that distribution matching in the product space yields a tighter generalization error bound than Euclidean-only matching, and reports experimental results on MNIST, CIFAR-10, CIFAR-100, Imagenette, ImageWoof, and ImageNet-100 showing consistent improvements over existing distribution-matching baselines.

**Compliance With Llm Reviewing Policy:**

Affirmed.

**Final Justification:**

The authors' reply to the key questions of my initial review appeared during the discussion period adequately address my main concerns. I am willing to raise my score accordingly.

**Key Questions For Authors:**

- Is there any analysis showing that the learned curvatures actually reflect the intrinsic geometry of the data manifold?
- Beyond accuracy improvements, does geometry-aware matching provide benefits unique to the dataset distillation setting, e.g., achieving comparable accuracy with fewer synthetic samples per class?
- The product space is restricted to constant-curvature components. Why were more general manifold learning approaches [5] not considered as alternatives?

## References

[1] Gu, A., Sala, F., Gunel, B., & Ré, C. (2018). Learning mixed-curvature representations in product spaces. International Conference on Learning Representations.
[2] Cho, S., Cho, S., Park, S., Lee, H., Lee, H., & Lee, M. (2023). Curve your attention: Mixed-curvature transformers for graph representation learning. arXiv preprint arXiv:2309.04082.
[3] Liu, H., Li, Y., Xing, T., Dalal, V., Li, L., He, J., & Wang, H. (2023). Dataset distillation via the Wasserstein metric. arXiv preprint arXiv:2311.18531.
[4] Cui, X., Qin, Y., Zhou, W., Li, H., & Li, H. (2025). OPTICAL: Leveraging optimal transport for contribution allocation in dataset distillation. Proceedings of the Computer Vision and Pattern Recognition Conference.
[5] Rozo, L., González-Duque, M., Jaquier, N., & Hauberg, S. (2025). Riemann $^ 2$: Learning Riemannian Submanifolds from Riemannian Data. arXiv preprint arXiv:2503.05540.

**Limitations:**

yes

**Strengths And Weaknesses:**

## Strengths

- The paper is well-written, easy to follow, and the motivation grounded in the manifold hypothesis is clearly articulated.
- The experimental evaluation is thorough, with consistent improvements demonstrated across diverse benchmarks, extensive ablation studies, and supplementary analyses covering cross-architecture transfer, computational cost, dimension allocation, backbone variations, and initialization sensitivity.

## Weaknesses

- The core methodological components, i.e., learnable curvature, learnable geometry weights, and optimal transport loss, are largely orthogonal to dataset distillation itself. Learnable curvature in product manifolds has been explored in prior work on non-Euclidean representation learning [1, 2], and OT-based losses have already been applied in the distillation literature [3, 4]. The contribution here amounts to importing these existing techniques into the distribution-matching pipeline for dataset distillation, rather than developing mechanisms that are specifically motivated by the unique challenges of distillation. The paper would be strengthened if it demonstrated benefits that are distinctive to the distillation setting.
- The theoretical analysis rests on the assumption that the data manifold $\mathcal{M}^*$ is a subset of the Euclidean–hyperbolic–spherical product space. Such assumption is unlikely to hold for any of the benchmark datasets used in the experiments, nor for most real-world data in general. Moreover, since Euclidean space is a special case of the product space, any expressiveness advantage is already trivially expected with the assumption.

---

> ### Author Rebuttal · Authors · 2026-03-31
>
> **Weakness 1**:Our contribution is not the introduction of individual components, but a new formulation of dataset distillation as geometry-aware manifold alignment. To our knowledge, this is the first work that explicitly connects distribution matching in DD with the manifold hypothesis and studies it in a product Riemannian space, which also provided a theoretical foundations.
>
> Our theory in section 4 shows that geometric distortion directly affects the generalization gap, and that product-space matching yields a tighter bound than Euclidean matching. This provides a distillation-specific motivation: under extreme compression, preserving intrinsic geometry is critical for maintaining distribution fidelity.
>
> We note that prior work such as hyperbolic dataset distillation introduces non-Euclidean geometry by performing distribution matching in a single hyperbolic space, thus capturing only one type of geometric structure.
> In contrast, our approach models data in a product manifold and aligns distributions across multiple geometries with learnable curvature and weights, enabling the model to capture heterogeneous structures in a unified framework.
>
> Based on this formulation, the product space, learnable curvature/weights, and OT loss naturally arise to better align the data manifold. Empirically, the gains are most significant in low-IPC settings, further supporting that geometry-aware modeling addresses a key challenge in dataset distillation.
>
> **Weakness 2**: We clarify that the assumption $\mathcal{M}^* \subset E \times H \times S$ is not meant to be a literal description of real data, but an analytical abstraction to study geometric distortion. Our theory does not require data to exactly lie in this space, but shows that when data exhibit mixed geometric structures, embedding them into a product space can reduce distortion compared to purely Euclidean embeddings (Theorem 4.1).
>
> Empirically, we observe such mixed structures in real datasets. For example, Fig. 1 in original paper shows that embeddings reveal hierarchical patterns in hyperbolic space and spherical concentration in spherical space, indicating that real data are not strictly Euclidean.
>
> Regarding the concern of trivial expressiveness, our improvement does not come from increased capacity, but from better geometric fidelity. Euclidean space is indeed a special case, but it introduces distortion when modeling hierarchical or angular structures, while the product space reduces this distortion, leading to improved distribution alignment.
>
> Moreover, when such geometric structures are weak, our method degenerates to Euclidean-like behavior via learnable weights, ensuring no performance degradation.
>
> **Question 1**: We provide empirical evidence that the learned curvatures are not arbitrary but reflect underlying data structure. In Appendix (Table 13), we show that different random initializations consistently converge to similar curvature and weight configurations, indicating that the learned geometry is data-driven rather than optimization noise.
>
> Moreover, the learned parameters vary across datasets and settings, suggesting that the model adapts to different intrinsic structures.
> This is also consistent with our observation in Fig. 1, where real data exhibit distinct geometric patterns.
>
> Finally, the performance gains are closely tied to these learned geometric parameters, indicating that they capture meaningful structural information.
>
> **Question 2**: Beyond absolute accuracy, our results demonstrate improved sample efficiency. For example, on MNIST, GeoDM with IPC=1 already outperforms NCFM with IPC=10, and GeoDM with IPC=10 surpasses NCFM with IPC=50 in Table 1. This shows that our method can achieve comparable or even better performance with significantly fewer synthetic samples.
>
> **Question 3**: Our method does not assume fixed constant-curvature spaces. Instead, we introduce learnable curvature parameters for the hyperbolic and spherical components, allowing the geometry to adapt to the data. In this sense, our model already breaks the fixed-geometry assumption while maintaining a structured representation.
>
> We believe the reviewer refers to more general manifold learning approaches with spatially varying curvature or fully learned manifold. While such models are more flexible, they typically come with significantly higher computational and optimization complexity, and are harder to integrate into the distribution matching framework.
>
> In contrast, our product space can be viewed as a principled basis that captures diverse geometric structures, while remaining tractable and theoretically analyzable (Theorem 4.1–4.2). This strikes a balance between expressiveness and stability, which is particularly important for dataset distillation.

---

> > ### Author Rebuttal · Reviewer_csNB · 2026-04-04
> >
> > The authors are thanked for their detailed response, and the demonstrated sample efficiency is acknowledged as impressive. Nevertheless, the concern regarding the gap between the theoretical framework and the experimental evaluation remains.
> >
> > The theory relies on the assumption that data lies in a product of constant curvature spaces, yet none of the datasets used in the experiments are known to satisfy this assumption. This raises the concern that the theoretical guarantees may rest on assumptions that are too strong to be practically relevant.
> >
> > Additionally, regarding the response to Q1, the authors argue that convergence points being consistent across seeds but varying across datasets indicates that the proposed method captures the underlying geometry of each dataset. This argument is not fully convincing. Such behavior, i.e., seed-invariant yet dataset-dependent convergence, could plausibly emerge even when the data does not reside in a product of constant curvature spaces, and thus does not, on its own, serve as evidence that the method is faithfully reflecting the geometric structure.
> >
> > For these reasons, the score remains unchanged.

---

> > > ### Author Response · Authors · 2026-04-06
> > >
> > > We thank the reviewer for the detailed feedback.
> > >
> > > Regarding the concern that our theory relies on the assumption that real data lie in a product of constant-curvature spaces, we clarify that this  is supported by our motivation. As shown in Fig. 1, standard datasets such as CIFAR-10 exhibit clear geometric patterns, indicating that real data are not purely Euclidean. This observation motivates our formulation.
> > >
> > > Moreover, our method consistently improves performance across different datasets, suggesting that product-space modeling captures additional geometric information beyond Euclidean embeddings. Importantly, even if the data do not strictly follow such geometric assumptions, the model can adaptively reduce to Euclidean behavior via learnable weights, ensuring no degradation.
> > >
> > > Regarding the concern about intrinsic curvatures, we emphasize that consistent convergence across different initializations already indicates that the learned curvature is data-driven rather than random. Moreover, we provide stronger empirical support. In particular, when we replace 50\% of CIFAR-10 with Gaussian noise (which removes structured information), the learned spherical and hyperbolic curvatures decrease significantly, which is listed in the answer to **Question 3 for Reviewer hvXa**. This demonstrates that the learned geometry adapts to the underlying data structure, providing evidence that it reflects meaningful properties rather than arbitrary optimization effects.
> > >
> > > Finally, we note that the other reviewers acknowledged the motivation and theoretical design, which further supports that our formulation is grounded in both theory and empirical observations.

---

### Decision · Program_Chairs · 2026-04-30

**Decision:**

Accept (regular)

**Comment:**

This paper proposes a geometry-aware distribution matching framework for dataset distillation. The core idea is to match distributions in a Cartesian product space comprising Euclidean, hyperbolic, and spherical manifolds, into which samples are embedded via a Riemannian convolutional neural network. The paper provides theoretical analysis arguing that distribution matching in this product space yields tighter generalization error bounds than purely Euclidean approaches.

All reviewers agree that the paper addresses a meaningful problem: the intuition that real data may exhibit heterogeneous geometric structure and that Euclidean embeddings alone may be insufficient is well-motivated and broadly appreciated. The paper is also considered globally well-written and the theoretical contributions were generally well-received.

The reviews raised two main concerns. First, the realism of the three-manifold assumption in practice. Second, experimental concerns about design and hyperparameter choices, including the CNN architecture, the dimensional split, and the learnable curvature design (partly due to the absence of reproducible code). Limited evaluation, focusing mainly on vision benchmarks without comparison against other "geometry-aware" distillation methods, was also criticized.

The authors provided a serious rebuttal addressing these points. Notably, the clarifications on design choices, architectures, and dimensions should be incorporated into the final version. The authors also extended their evaluation to more challenging datasets and included comparisons against hyperbolic distillation methods. Finally, they convincingly argue that the product space is an analytical abstraction rather than a literal description of real data, and that it helps reduce geometric distortion under mixed geometric structures.

Following the rebuttal, the previously most critical reviewer indicated that their main concerns had been adequately addressed and expressed willingness to raise their score. Given the broad appreciation for the motivation, the quality of the experiments, and the theoretical contributions, I recommend acceptance.